# Loss of *Dnah5* Downregulates *Dync1h1* Expression, Causing Cortical Development Disorders and Congenital Hydrocephalus

**DOI:** 10.3390/cells13221882

**Published:** 2024-11-14

**Authors:** Koichiro Sakamoto, Masakazu Miyajima, Madoka Nakajima, Ikuko Ogino, Kou Horikoshi, Ryo Miyahara, Kaito Kawamura, Kostadin Karagiozov, Chihiro Kamohara, Eri Nakamura, Nobuhiro Tada, Akihide Kondo

**Affiliations:** 1Department of Neurosurgery, Juntendo University Graduate School of Medicine, 2-1-1 Hongo, Bunkyo-ku, Tokyo 113-8421, Japan; madoka66@juntendo.ac.jp (M.N.); i-ogino@juntendo.ac.jp (I.O.); k-horikoshi@juntendo.ac.jp (K.H.); r-miyahara@juntendo.ac.jp (R.M.); k-kawamu@juntendo.ac.jp (K.K.); kostadinkaragiozov@yahoo.com (K.K.); c-kamohara@juntendo.ac.jp (C.K.); knd-aki@juntendo.ac.jp (A.K.); 2Department of Neurosurgery, Juntendo Tokyo Koto Geriatric Medical Center, 3-3-20 Shinsuna, Koto-ku, Tokyo 136-0075, Japan; 3Department of Genetic Analysis Model Laboratory, Juntendo University Graduate School of Medicine, Hongo Bunkyo-ku, Tokyo 113-8421, Japan; enakamur@juntendo.ac.jp (E.N.); ntada@juntendo.ac.jp (N.T.)

**Keywords:** *Dnah5*, motile cilia, primary cilia, neurogenesis, cytoplasmic dynein

## Abstract

*Dnah5* is associated with primary ciliary dyskinesia in humans. *Dnah5*-knockout (*Dnah5*−/− mice develop acute hydrocephalus shortly after birth owing to impaired ciliary motility and cerebrospinal fluid (CSF) stagnation. In contrast to chronic adult-onset hydrocephalus observed in other models, this rapid ventricular enlargement indicates additional factors beyond CSF stagnation. Herein, we investigated the contributors to rapid ventricular enlargement in congenital hydrocephalus. *Dnah5*−/− mice were generated using CRISPR/Cas9. The expression of dynein, N-cadherin, and nestin in the cerebral cortex was assessed using microarrays and immunostaining. Real-time PCR and Western blotting were performed for gene and protein quantification, respectively. All *Dnah5*−/− mice developed hydrocephalus, confirmed by electron microscopy, indicating the absence of axonemal outer dynein arms. Ventricular enlargement occurred rapidly, with a 25% reduction in the number of mature neurons in the motor cortex. *Dync1h1* expression was decreased, while cytoplasmic dynein levels were 56.3% lower. Levels of nestin and N-cadherin in the lateral ventricular walls decreased by 31.7% and 33.3%, respectively. Reduced cytoplasmic dynein disrupts neurogenesis and axonal growth and reduces neuron cortical density. Hydrocephalus in *Dnah5*−/− mice may result from cortical maldevelopment due to cytoplasmic dynein deficiency, further exacerbating ventricular enlargement due to CSF stagnation caused by impaired motile ciliary function.

## 1. Introduction

Congenital hydrocephalus is a common anomaly worldwide. Although the precise mechanisms underlying the various types of congenital hydrocephalus are not fully understood, a significant number of genetic anomalies associated with this condition have been identified, including primary ciliary dyskinesia (PCD) and hydrocephalus complicated by other ciliopathies [1]. Congenital hydrocephalus often leads to impaired cortical development, even with surgical intervention in the early postnatal stages, resulting in a high incidence of moderate-to-severe developmental delays and challenges with social integration [2,3]. Mutations in ciliary-related genes have been extensively reported and are considered to be linked to congenital hydrocephalus [1,4,5]. However, as summarized in our review in 2021, the association between ciliary motility disorders and hydrocephalus in humans remains limited, with only approximately a dozen reports to date [6]. Several studies using animal models have explored specific genes related to ciliary motility, revealing abnormalities in ciliary length or number that affect ciliary motility and cerebrospinal fluid (CSF) flow, subsequently inducing ventricular enlargement [7,8]. In humans, genes such as *Foxj1*, *Cfap43*, and *Cwh43*, which have been identified as causes of PCD, were shown to be linked to adult-onset chronic hydrocephalus, demonstrating phenotypes similar to those observed in animal models [9,10,11,12]. Knockout of the dynein axonemal heavy chain 5 (*Dnah5*) gene, one of the most frequently mutated genes in human PCD, was found to result in rapid progression of hydrocephalus in mice, leading to death within approximately one month [13,14]; however, no clear association between *Dnah5* and the onset of hydrocephalus in humans has been established to date.

Mutations in the causative genes lead to various abnormalities in the ciliary structures, resulting in ciliary motility disorders and the development of hydrocephalus with diverse outcomes. Moreover, the same genetic mutation may lead to significantly different phenotypes between humans and animal models [9,10,11,12,13,14,15,16,17]. Previous reports have noted that in both humans and animal models, some cases of ciliary motility disorders in PCD do not lead to the development of hydrocephalus. Although ciliary motility disorders have been implicated in the onset of hydrocephalus, they have been considered insufficient as the sole factor. In this study, we aimed to elucidate factors other than ciliary motility disorders involved in the onset of hydrocephalus.

## 2. Methods

### 2.1. Animals

All experimental animals were group-housed with 2–5 animals per cage and maintained in a temperature- and humidity-controlled facility (23 ± 1 °C, 55 ± 5% humidity, 12 h light/12 h dark) at the animal care facility in the Center for Experimental Medicine of Juntendo University, Japan. All procedures involving animals received approval from the Ethics Review Committee for Animal Experimentation of the Juntendo University School of Medicine (approval 1337) and were performed in accordance with the principles of laboratory animal care outlined by the National Institutes of Health.

After measuring the body weight of all animals, deep anesthesia was induced using a mixed anesthetic agent (0.3 mg/kg of medetomidine, 4.0 mg/kg of midazolam, and 5.0 mg/kg of butorphanol) administered intraperitoneally, followed by decapitation for brain dissection, following the research protocol. Each brain sample was immediately stored in RNAlater^®^ solution (AM7021; Thermo Fisher Scientific, Waltham, MA, USA) to preserve the material for subsequent DNA, RNA, and protein extractions.

We have provided a timeline tracking knockout mice grouped by postnatal age, to better depict our research, and have provided a summary of our thought processes in a flowchart (Figure A1).

### 2.2. Generation of Dnah5−/− Mice

*Dnah5*-knockout (*Dnah5*−/−) mice were generated in our facility using the CRISPR/Cas9 system [18,19]. Cas9 protein (Takara Bio Inc., Shiga, Japan) and sgRNA were microinjected into the cytoplasm of fertilized one-cell eggs at the pronucleus stage in C57BL/6J female mice (Charles River Laboratories Japan, Inc., Yokohama, Japan).

Mutations were evaluated using polymerase chain reaction (PCR) followed by T7E1 enzyme assays. The sequence of sgRNA used for *Dnah5*−/− mouse generation was 5′-CCTGTTTGGACCTGAACAAACCA-3′ (chromosome 15, GRCm38. p4). Genomic DNA was extracted from the tail chips of *Dnah5*−/− mice using a DNeasy blood and tissue kit (Qiagen, Hilden, Germany) following the manufacturer’s instructions. PCR conditions were 98 °C for 10 s, 60 °C for 30 s, and 72 °C for 1 min for 30 (1st PCR) and 35 cycles (2nd PCR). The primer sequences used for genotyping of *Dnah5*−/− mice were as follows: F_1st: 5′-GGCAACGGAGGTCAGCAAT-3′; R_1st: 5′-AGAAGCAGGCATCATCATCAA-3′; F_2nd: 5′-GGTCCCATTAAGCTGCCTGCTA-3′; and R_2nd: 5′-ACAAGTACCATCATTCAACCTGGAG-3 The annealed PCR products were treated with T7 endonuclease to evaluate mismatches at the target site. The PCR products were column-purified using Amicon Ultra 0.5 mL 30 kDa (Merck KGaA, Darmstadt, Germany) and sequenced to determine mutations. Consequently, a single strain of *Dnah5*−/− mice was established.

### 2.3. Genotyping of Dnah5−/− Mice

To verify the genotypes of the mice, PCR amplification of a 107 base-pair (bp) fragment was performed using a PyroMark PCR kit (978703; Qiagen, Hilden, Germany) with 10 ng mouse DNA for 4% gel loading and pyrosequencing. PCR and sequencing primers for *Dnah5* were as follows: forward, 5′-ATG GGC GGC ATG ACT ATC-3′; reverse, 5′-CAC AAA GCT TCC TAC CTG ATT ACC-3′; sequence, 5′-TCT GTT TGC AAT TGT AGC-3′; and dispensation order 5′-GTC ATG TGA CT-3′.

### 2.4. Reverse-Transcription PCR (RT-PCR)

Total RNA was extracted from 12 to 20 mg of cerebral ventricle and cortex tissues of three-week-old mice using an RNeasy Mini Kit (Qiagen). cDNA was synthesized using SuperScript™ⅣVILO™Master Mix (Thermo Fisher Scientific), and PCR was performed using a ProFlex^TM^ PCR system (Thermo Fisher Scientific). *GAPDH* was used as the reference gene for *Dnah5* expression.

The PCR primers were as follows: *Dnah5* (XM_006520014, 423 bp) forward, 5′-CTT CAC ACC GGA CAA CAA GC-3′; reverse, 5′-TCA CGA ACA CCT TGG GCT TT-3’; *GAPDH* (XM_017321385, 1164 bp) forward, 5′-TCA CCA CCA TGG AGA AGG C-3′; reverse, 5′-GCT AAG CAT TGG TGG TGC A-3′.

### 2.5. Hematoxylin–Eosin (HE) Staining

Brains of wild-type and *Dnah5*−/− mice collected on days zero, three, five, eight, and ten were fixed in 4% paraformaldehyde fixative (33111; Muto Pure Chemicals Co., Tokyo, Japan) for at least 1 week, embedded in paraffin, and cut into 5 μm sections. Sections were stained with 3,3′-diaminobenzidine, counterstained with Mayer’s hematoxylin, dehydrated, cleared, and mounted. The sections were viewed under an E800 microscope (Nikon, Tokyo, Japan) and images were captured with an AxioCam 506 color digital camera using AxioVison Rel version 4.7.2.0 image-processing software (Carl Zeiss Microimaging GmbH, Jena, Germany).

### 2.6. Ventricular Injection of Dye (DiI)

Brains of *Dnah5*−/− mice on day four were fixed in a stereotaxic apparatus (Narishige Scientific Instrument Laboratory, Tokyo, Japan), while a small burr hole was drilled into the cranium on the right side 1 mm lateral to the bregma. The needle of a microsyringe (MS-E05; Ito Corporation, Shizuoka, Japan) was slowly inserted into the lateral ventricle 1.5 mm from the brain surface through this hole, allowing the injection of 1 μL of Cell Tracker^TM^ CM-DiI dye (C7001; Thermo Fisher Scientific) at a rate of 2 μL/min. Ten minutes after injection, mouse brains were harvested, fixed in 4% paraformaldehyde for 72 h, and then cut into 30 μm coronal cryosections (CM3050 S, Leica Microsystems GmbH, Wetzler, Germany).

### 2.7. Scanning Electron Microscopy (SEM) and Transmission Electron Microscopy (TEM)

The brains of wild-type and *Dnah5*−/− mice were perfused with 4% paraformaldehyde. The materials were immersed in a 2.5% glutaraldehyde solution after being cut into smaller fragments for TEM (1 × 2 × 2 mm^3^) and SEM (3 × 3× 3 mm^3^). Sections were then washed in phosphate-buffered saline (PBS; AJ9P003; TaKaRa, Shiga, Japan), post-fixed in 2% osmium tetroxide for 2 h at 4 °C, and dehydrated with graded concentrations of ethanol. For TEM, samples were placed in resin for 4 days at 60 °C. The ultrathin sections were observed under an HT7700 electron microscope (Hitachi High Technologies, Tokyo, Japan).

For SEM analysis, the samples were freeze-dried. The gold palladium-coated tissues were scanned using an S-4800 scanning electron microscope (Hitachi High Technologies, Tokyo, Japan).

### 2.8. Confirmation of Ciliary Motility

To confirm ciliary movement, we removed the brains of the mice on day 10, and created 1 mm brain slices. Subsequently, 10 μm microbeads (Polybeads^®^ Polystyrene Black Dyed 10.0 Micron Microspheres, 24294-2, Polysciences Inc., Warrington, PA, USA) were diluted 5 times with PBS, and 1 μL of this diluted solution was placed into the lateral ventricle after removing the choroid plexus. The movement of the microbeads was recorded for 5 min at a rate of 6 fps using a video microscope (DMI 3000 B, Leica) and Leica Application Suite, 4.2.0 software. The bead trajectories were traced using ImageJ 2.1.0 and TrackMate 6.0.1.

### 2.9. Microarray Analysis

Gene expression microarray analysis was performed using the hemispheric brain tissue of wild-type and *Dnah5*−/− mice on day three. The RNA quality and integrity were measured using a 4150TapeSation (Agilent Technologies, Santa Clara, CA, USA). For 250 ng of total RNA, a WT Expression Kit and GeneChip^TM^Mouse Exon 1.0 ST Array (Affymetrix, Thermo Fisher Scientific) were used, following the manufacturer’s protocol. We used Transriptome Analysis Console (TAC) software (4.0.2.15) and the Robust Multichip Average (RMA) algorithm for normalization. Data from the comparative analysis over and below log2 fold-change cut-off values were entered into Ingenuity Pathway Analysis (IPA, Fall 2024 version 1.0) software (Qiagen, Hilden, Germany). Disease and functional analyses were performed using IPA.

### 2.10. Real-Time Quantitative PCR

Real-time quantitative PCR was conducted using brain cortices of three-day-old *Dnah5*−/− mice. Total RNA (250 ng) was converted into single-stranded cDNA using SuperScript IV VILO™ (SSIV VILO) Master Mix (11756050, Invitrogen, Thermo Fisher Scientific). An ABI 7500 real-time PCR system (Applied Biosystems, Thermo Fisher Scientific) and TaqMan^®^ gene expression assays (Applied Biosystems, Thermo Fisher Scientific) were applied to quantify gene expression, according to the instructions from the manufacturer. The assay IDs are listed in Table A1. Target gene expression was standardized to that of *Actb*. The presence of a single PCR amplicon was confirmed using melting curve analysis. The expression of each gene in each sample was analyzed in triplicate. Real-time quantitative PCR results were quantified using the 2^−ΔΔCt^ method.

### 2.11. Immunofluorescence Staining

Brains of three-day-old wild-type and *Dnah5*−/− mice were removed and post-fixed in 4% paraformaldehyde in 0.01 M phosphate buffer (pH 7.2). Paraffin-embedded sections (5 μm) were blocked with 5× SEA BLOCK™ blocking buffer (37527; Thermo Fisher Scientific) and 1% donkey serum in PBS for 30 min, incubated in primary antibody overnight at 4 °C, and then in secondary antibodies for 60 min at room temperature. The primary and secondary antibodies used are listed in Table A2.

Nuclei in all sections were examined under a confocal scanning microscope (TCS-SP5; Leica Microsystems) using Leica Application Suite X 3.4.2.18368 (Leica Microsystems, GmbH, Wetzlar, Germany).

### 2.12. Cell Count

The numbers of nuclei of mature neurons were counted in all layers of the selected three-day-old wild-type and *Dnah5*−/− mouse cerebral motor cortices stained with Neuro Trace. Eight slices were selected for each *Dnah5*−/− and wild-type mouse, and the width was set to 100 µm.

### 2.13. Western Blotting Analysis

Proteins were extracted from the brain cortices of *Dnah5*−/− mice on day three and lysed in 200 µL of lysis buffer (N-PER; Thermo Fischer Scientific) containing a protease inhibitor cocktail (cOmplete ULTRA Mini EDTA-free EASYpack; Roche, Basel, Switzerland). Lysates were segregated by centrifugation at 20,000× *g* at 4 °C for 15 min, and protein concentrations of the resultant supernatants were determined using BCA protein assay kits (Thermo Fischer Scientific). Subsequently, protein aliquots (10 µg for dynein and 20 µg for N-cadherin and nestin) were heated at 70 °C for 10 min in NuPAGE^®^ LDS Sample Buffer (NP0008; Invitrogen). Samples were electrophoresed on 3–8% NuPAGE^®^ Tris-Acetate Mini Gel by NuPAGE^®^ Tris-Acetate SDS Running Buffer (20×) (LA0041; Invitrogen) and then transferred to a polyvinylidene fluoride membrane. The primary antibodies used are listed in Table A2. Chemiluminescent signals were detected using a Western Breeze Kit (WB7106; Invitrogen). Immunoreactive bands were detected using ImageLab version 4.1 software (Bio-Rad Laboratories, Hercules, CA, USA).

### 2.14. Statistical Analysis

All experimental results are presented as means ± standard deviation. Data distribution was evaluated graphically using histograms and Q–Q plots. The Shapiro–Wilk test was applied to assess the normality of the distributions. For real-time quantitative PCR performed in at least five independent experiments, significant differences among groups were determined using Welch’s *t*-test. Statistical differences in the number of neurons in the motor cortex between wild-type and *Dnah5*−/− mice were assessed using the Mann–Whitney U test. All calculations were performed using IBM SPSS Statistics 25, with statistical significance set at *p* < 0.05.

## 3. Results

### 3.1. Generation of Dnah5−/− Mice

*Dnah5*−/− mice were generated to investigate the effect of the *Dnah5* gene on the brain tissue. DNA sequencing confirmed a 4 bp deletion in the targeted exon 2 of *Dnah5* on chromosome 15 in the *Dnah5*−/− mice (Figure 1A). Consequently, a frameshift mutation occurred, resulting in the 25th codon becoming a TGA stop codon. RT-PCR results also confirmed the absence of a 500 bp band in *Dnah5*−/− mice, which was detected in wild-type mice (Figure 1B). The pyrosequencing results also confirmed the genotype of the generated *Dnah5*−/− mice, further confirming the 4 bp deletion (Figure 1C). Hydrocephalus and situs inversus occurred in all and 26% of homozygous individuals among the *Dnah5*−/− mice (Figure 1D). A comparative analysis of body weight changes at zero, three, five, eight, and ten days old revealed a significant difference on day eight, with no significant differences observed until day five (Figure A2). Comparison of the head morphologies of five-day-old mice revealed a distinct round deformation in *Dnah5*−/− mice compared to that in wild-type mice (Figure 1D). Examination of the ventricular morphology of *Dnah5*−/− mice using coronal sections stained with HE revealed that these mice had already displayed enlargement of the lateral ventricles early after birth, which became more pronounced on the subsequent days (Figure 1E).

### 3.2. Ciliary Structure and CSF Flow

Transmission electron microscopy (TEM) analysis of the cilia cross section in *Dnah5*−/− mice on day three revealed that the outer dynein arm structure of the peripheral doublet microtubules was completely absent (Figure 2A, right top). The overall morphology of the motile cilia, based on SEM images, further revealed uniformly formed ciliary bundles that extended into the ventricular cavity in wild-type mice. In contrast, *Dnah5*−/− mice exhibited an uneven morphology that appeared to collapse towards the ventricular wall (Figure 2A, right bottom). An experiment involving dye injection by puncturing the right lateral ventricle with a microneedle and injecting DiI (a fluorescent dye) was conducted using *Dnah5*−/− mice on day four (Figure 2B). The results revealed no occlusion of the cerebral aqueduct, despite ventricular enlargement. Therefore, enlargement of the ventricles was observed before occlusion of the cerebral aqueduct (Figure 2B). To confirm the ependymal flow by motile cilia, coronal sections were obtained immediately following extraction from live mice. Microbeads were placed in the lateral ventricular cavity of the slice, and the flow of microbeads in PBS was recorded and tracked. In wild-type mice, a consistent vortical flow was observed, whereas no flow of cerebrospinal fluid was detected in *Dnah5*−/− mice (Figure 2C).

### 3.3. Effect of Dnah5 Gene Deficiency on the Cortex

Coronal sections of the cerebral cortex of *Dnah5*−/− and wild-type mice were examined on day three. Immunostaining of the motor cortex with Neuro Trace revealed that mature neurons were densely packed and arranged radially in the cerebral cortex of wild-type mice, whereas an uneven arrangement was observed in *Dnah5*−/− mice (Figure 3A). The nuclei of mature neurons in the motor cortex were further counted throughout the entire cortical layer in 100 μm intervals using Neuro Trace (Figure 3A, left). An approximately 25% reduction in cell count was observed in *Dnah5*−/− mice compared to wild-type mice (Figure 3A, right). Microarray analysis was performed using one hemisphere of the *Dnah5*−/− mouse brain. A total of 1052 of the 2534 genes selected by TAC were analyzed using Ingenuity^®^ Pathway Analysis (IPA) for the “Disease and Function” category. Upon ranking based on Fisher’s exact test p-values, the top category was found to be the “Development of neural cells” (Table 1). Significant downregulation of *NeuroD6* and *NeuroD2*, which is involved in the same pathway as *NeuroD6*, in *Dnah5*−/− mice was also observed [20] (Table 2). Real-time quantitative PCR was also conducted for the *NeuroD6*, *NeuroD2*, and *Ascl1* genes, which are all involved in the same pathway. The results indicated decreased expression compared to the control, with relative values of 0.357, 0.445, and 0.421 for *NeuroD6*, *NeuroD2*, and *Ascl1*, respectively. This decrease was attributable to *Dnah5* knockout. The expression of *Dync1h1*, a gene involved in the heavy-chain structure of cytoplasmic dynein, also decreased (Table 3). Real-time quantitative PCR also confirmed a substantial decrease in the expression of *Dync1h1* to 0.326 relative to the control.

Observation of dynein in the cerebral cortex revealed that in wild-type mice, the expression of the anti-dynein antibody was radial in direction, coinciding with the cortical neuronal axons. In contrast, expression was low in *Dnah5*−/− mice (Figure 3B). Western blotting confirmed a 60% decrease in dynein expression (Figure 3C).

### 3.4. Effect on the Ependymal and Subependymal Zone Layers

HE staining was performed to observe the ependymal layer of the lateral ventricles in wild-type and *Dnah5*−/− mice at three days of age (Figure 4A). The cell densities in the ependymal and subependymal zone layers lining the lateral ventricle wall in *Dnah5*−/− mice were sparse, and the layers were coarser than those in wild-type mice. Immunostaining of the ependymal and subependymal zone layers of the lateral ventricle wall with Neuro Trace revealed a reduced cell density in *Dnah5*−/− mice (Figure 4B). Nestin antibody staining in the same region was reduced in the ependymal and subependymal layers of *Dnah5*−/− mice compared to wild-type mice. N-cadherin antibody staining was further reduced in *Dnah5*−/− mice (Figure 4C). Western blotting confirmed a 31.7% decrease in nestin and 33.3% decrease in N-cadherin protein expression levels (Figure 4D).

## 4. Discussion

### 4.1. Genes Associated with Cilia Motility

A previous case report documented the co-occurrence of hydrocephalus and developmental disorders in three generations of a Jordanian family with autosomal recessive PCD [21]. Additionally, mutations in the genes *Dnah2* and *Dnah14*, which contribute to motile cilia structure, have been associated with primary microcephaly, neurodevelopmental disorders, and other conditions such as seizures [22,23]. *Dnah2* and *Dnah14* are essential for the inner dynein arm structure within the peripheral doublet microtubules of motile cilia [24]. We previously reported that mutations in *Dnah14* are linked to the development of chronic hydrocephalus in humans [25].

Hydrocephalus, a common hallmark of PCD, has been linked to PCD in genetically modified animals, particularly in mouse models, in several studies [26]. Sawamoto et al. used Tg737 mutant mice to demonstrate the involvement of motile cilia in neuronal migration [27]. However, a direct causal relationship between the cilia-related genes and brain formation has not yet been established. Therefore, we used a knockout mouse model of *Dnah5*, a representative causative gene for PCD, and focused on (i) changes in the cerebral cortex immediately after birth and (ii) changes in the ventricular wall before the onset of hydrocephalus to identify the mechanisms underlying hydrocephalus.

Ibanez et al. created a mouse model that stopped at exon 16, as within exon 79 on chromosome 15, there are six P-loops responsible for ATP hydrolysis and a microtubule-binding site from exon 17 onwards, which are involved in ciliary motility. Approximately 20% of these mice developed hydrocephalus, with aqueductal obstruction occurring at 3–5 days of age, followed by rapid ventricular enlargement after day 5 [13,14]. Herein, we created a mouse model using the CRISPR/Cas9 technology, resulting in the complete absence of the outer dynein arms and 100% incidence of hydrocephalus. Our findings revealed mild ventricular enlargement from birth, with simultaneous impairment of cerebral cortex formation. Hence, *Dnah5* may affect not only the motility of ependymal cilia but also the formation and development of the cerebral cortex.

### 4.2. The Effect on Cerebral Cortex Formation in Dnah5−/− Mice

Mild ventricular enlargement was observed early after birth in *Dnah5*−/− mice. However, from day five and onwards, weight gain deteriorated upon completion of aqueductal obstruction and obvious ventricular enlargement was observed. Compared to wild-type mice, no significant weight difference was observed prior to aqueductal obstruction. Therefore, experiments were conducted using mouse brains from day three, when the effects of eventual cortical compression due to ventricular enlargement and metabolic disturbances were less pronounced [28,29]. In the cerebral cortex of *Dnah5*−/− mice, a reduction in the cell density of mature neurons was observed on day three, as evidenced by Neuro Trace staining.

In both microarray and real-time quantitative PCR analyses of *Dnah5*−/− mice, a decrease in expression of *NeuroD* family members was observed. The *NeuroD* family comprises closely related proteins, including *NeuroD1*, *D2*, *D4*, and *D6*, which play essential roles as basic helix–loop–helix (bHLH) genes in regulating neurogenesis. Among these, *NeuroD2* and *D6* are crucial factors guiding axon navigation and regulating the axon tracts in the mouse cerebral cortex [30]. Studies of a double-mutant mouse model of *NeuroD2* and *D6* have revealed a decrease in functional glutamatergic synapses in the cerebral cortex, resulting in a reduced excitatory cortical network [31]. The observed reduction in the expression of *NeuroD* family members in the genetic analysis of *Dnah5*−/− mice suggests a potential impact on neuronal migration and proliferation. Gene mutation studies in patients with cortical formation abnormalities have further identified strong associations with genes, including *Dcx*, *Lis1*, *Tuba1a*, *Flna*, *Akt3*, *Pik3ca*, *Dync1h1*, *Kif5c*, *Kif7*, *Kif1a*, and *Kif26a* [32]. These genes play crucial roles in microtubule cytoskeleton formation within axons during cerebral cortex development. Mutations in *Dync1h1*, which encodes the heavy chain of cytoplasmic dynein, have further been reported to affect cortical development, leading to conditions such as microcephaly and abnormalities in brain structures, including nodular heterotopia, hypoplasia of the corpus callosum, and dysmorphism of the basal ganglia [33,34,35].

In the *Dnah5*−/− mouse model, reduced expression of *Dync1h1* led to the impairment of cytoplasmic dynein. Cytoplasmic dynein and kinesin play structural roles in the axonal microtubules of neurons and neuronal cell movement [36,37]. Decreased *Dync1h1* expression is involved in reducing cytoplasmic dynein, thereby interfering with neuronal migration and cerebral cortex formation.

### 4.3. Effect on the Structure of the Ventricular Wall in Dnah5−/− Mice

In addition to the role of cytoplasmic dynein in nuclear migration within neuronal axons in the cerebral cortex, it is also known to mediate material transport along the ciliary axoneme. To elucidate the effect of decreased cytoplasmic dynein on primary cilia, we investigated the cilia, ependymal cells, and subventricular zone layers. Cilia can be categorized into both motile and non-motile types, with motile cilia being further categorized into two subtypes with different structures: 9 + 2-structure cilia with central and peripheral microtubules involved in ciliary movement and 9 + 0-structure nodal cilia responsible for left–right axis determination [6,38]. Cytoplasmic dynein is a crucial protein for intracellular material transport in primary cilia [39].

Ependymal cells form a multiciliated monolayer of columnar epithelium covering the ventricular wall surface, beneath which lies the subventricular zone (SVZ). The SVZ detects various signals through primary cilia, which regulate cell division in the ventricular wall, impacting not only neuronal migration but also cerebrospinal fluid (CSF) dynamics [38,40,41,42,43,44,45,46,47]. Although primary cilia do not directly propel CSF movement, they relay information from CSF flow to nearby cells, adjusting the expression of molecules involved in regulating CSF flow. Primary cilia detect growth factors and chemical signals in the CSF, modulating the differentiation of neurons and glial cells according to the concentration gradients of these signals [27]. Moreover, primary cilia detect signaling molecules such as sonic hedgehog and Wnt, which contribute to regulation of gene expression, control of neuronal proliferation, differentiation, and migration. Dysfunction of primary cilia disrupts these essential signaling pathways, leading to abnormal neuronal migration and developmental disorders [48,49].

Abnormalities in primary ciliary function have further been linked to conditions such as Joubert, Meckel–Gruber, and Bardet–Biedl syndromes, highlighting their role in brain development and associated cognitive functions [50,51,52,53,54].

Radial glia, which extend primary cilia towards the ventricular surface, further play a central role in the proliferation of neural cells, and newly generated neural cells from this region migrate radially to form the cerebral parenchyma [41].

In *Dnah5*−/− mice, a decrease in cell density was observed in the ependymal and SVZ layers on day three, indicating reduced neurogenesis in these regions.

Cadherin is a cell adhesion molecule abundant in radial glia that plays a crucial role in the maturation of neural cells [55]. N-cadherin is a protein that mediates neuronal migration and adhesion. Loss of N-cadherin in radial glia has been shown to hinder cortical development, including cell proliferation, differentiation, and migration [56,57,58]. The observed reduction in *Dync1h1* expression leads to primary cilia dysfunction, disrupting the localization and expression of N-cadherin. Consequently, the migration of neural progenitor cells to appropriate locations and intercellular adhesion are compromised, thereby impairing normal formation of the ventricular wall and cerebral mantle.

Nestin is a cytoskeletal protein classified as an intermediate filament specific to neural stem cells and is a marker of these cells. Reduced intracellular material transport via primary cilia due to decreased *Dync1h1* expression disrupts the cytoskeletal structure, leading to reduced nestin expression. As nestin provides the structural support necessary for neural progenitor cell maintenance and division, a decrease in its expression is directly linked to the reduction and delayed development of neural progenitor cells.

Cytoplasmic dynein is crucial for the intra-flagellar transporter (IFT) of cilia, contributing to the maintenance of primary and motile cilia function [36,37,59,60,61]. The proper functioning of cytoplasmic dynein and kinesin in the IFT is also essential for maintaining cellular homeostasis. Downregulation of *Dync1h1* in *Dnah5*−/− mice resulted in the impairment of cytoplasmic dynein function, leading to the dysfunction of primary cilia, indicating a decrease in cell density in the ventricular wall due to impaired neurogenesis. Primary cilia control the concentration gradient of signals that regulate neural cell migration. The normal beating motion of motile cilia is crucial for this regulation [27]. In *Dnah5*−/− mice, significantly deficient ciliary motion disrupts the proper mixing of the CSF, and primary cilia dysfunction hinders signal detection from the ventricular side. Consequently, neurogenesis in the radial glia is impaired, leading to a decrease in cell density in the ependymal and SVZ layers.

In summary, *Dnah5* deficiency reduces *Dync1h1* expression, which in turn affects cytoplasmic dynein function in primary cilia, indirectly contributing to decreased nestin and N-cadherin expression. This association with reduced neural progenitor cells in the ventricular ependymal layer induced by *Dnah5*−/− suggests that migration deficits due to dysfunctional primary cilia and impaired neurogenesis result in cortical malformations. Here, we report for the first time that the impairment of neuronal migration causes cortical malformations due to dysfunctional primary cilia and compromised neurogenesis. We hypothesized that brain malformations and progressive hydrocephalus result in severe disabilities during the prenatal period in humans, potentially leading to a presumed non-viable birth. The genes examined in our experiments (*Dnah5*, *Dync1h1*, *Nes* (Nestin), and *CDH2* (N-cadherin)) were analyzed using IPA software, allowing us to construct the molecular signaling pathways shown in Figure A3. Although these results are not based on direct validation using *Dnah5*−/− mice, they suggest potential direct or indirect associations.

### 4.4. Considerations for Future Applications in Congenital Hydrocephalus Treatment

This study lays the background for several critical contributions to future treatment strategies for congenital hydrocephalus. In particular, elucidating the mechanisms of hydrocephalus onset due to *Dnah5*−/− offers valuable insights that may inform not only traditional therapies but also new approaches and therapeutic targets.

First, we propose focusing on *Dync1h1* and dynein function as potential therapeutic targets. Our research indicates that *Dnah5* deficiency-induced downregulation of *Dync1h1* is involved in cortical development abnormalities and hydrocephalus progression. Thus, molecular or gene therapies targeting *Dync1h1* or cytoplasmic dynein represent promising approaches. For example, therapies aimed at promoting *Dync1h1* expression or enhancing dynein function through drug use or transfection methods could potentially support cortical development and prevent the depletion of neural progenitor cells.

Second, attention could be directed toward restoring CSF flow through improving ciliary motility. This study has demonstrated that impaired ciliary motility contributes to CSF flow stasis, exacerbating hydrocephalus development. Consequently, therapeutic strategies aimed at enhancing ciliary function, such as drug use or gene repair technologies that promote ciliary motility, might offer a new approach for controlling hydrocephalus progression.

Third, we highlight the potential for early diagnosis and prevention of cortical developmental disorders. Given that aberrant expression of *Dnah5* and *Dync1h1* may lead to early-stage cortical developmental abnormalities, these genes could be applied as diagnostic markers. Early diagnosis would facilitate pre- or early postnatal therapeutic intervention, potentially preventing hydrocephalus and neurodevelopmental disorders.

In this manner, by clarifying how *Dnah5* and *Dync1h1* contribute to the causes and progression of congenital hydrocephalus, this research holds promise for identifying novel therapeutic targets, developing early diagnostic tools, and building a foundation for more effective personalized medicine.

### 4.5. Limitations

The primary limitation of this study is that we did not directly investigate primary cilia in the ependymal cell layer or their neuronal migration. Therefore, the mechanisms through which *Dnah5* affects cytoplasmic dynein and primary cilia remain unclear. Future studies should investigate primary cilia and neuronal migration to clarify these mechanisms.

## 5. Conclusions

In this study, we generated a genetically modified mouse model with knockout of *Dnah5*, a representative gene associated with PCD. Based on investigations of this model, we proposed mechanisms to explain the observed cortical malformations. Hydrocephalus in the *Dnah5*−/− mouse model was found to arise from the stagnation of CSF due to impaired motile ciliary function and cortical malformations due to cytoplasmic dynein deficiency.

## Figures and Tables

**Figure 1 cells-13-01882-f001:**
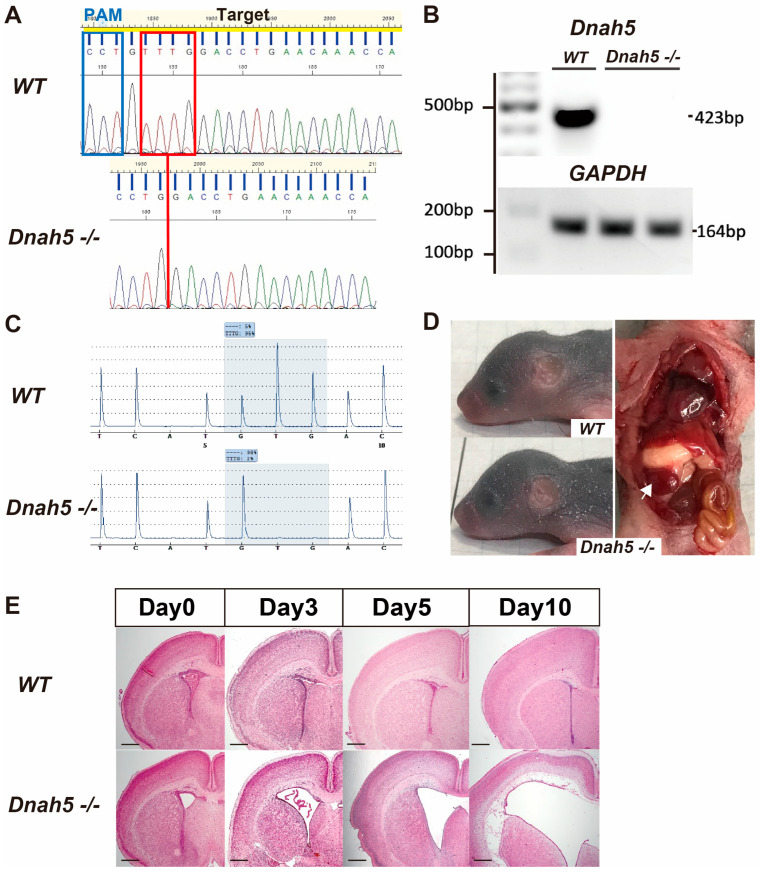
Generation of *Dnah5*−/− mice. (**A**) Schematic diagram of the *Dnah5* gene locus. The sgRNA sequence is underlined in yellow. The top row shows the wild-type gene sequence. The bottom row shows the gene sequence in the generated *Dnah5*−/− mice, revealing the deletion of 4 bp enclosed within a red box targeting exon 2 on chromosome 15, leading to a frameshift mutation. (sgRNA = single guide RNA) (**B**) Results of real-time quantitative PCR. The absence of the *Dnah5* gene was confirmed (**C**) The pattern of the 4 bp deletion in *Dnah5*−/− mice compared to wild-type mice was demonstrated using pyrosequencing. (**D**) The head shapes of mice on day five. The top image shows a representative wild-type mouse, while the bottom image shows a representative *Dnah5*−/− mouse, demonstrating changes in the head shape characteristic of hydrocephalus model mice. In the adjacent image, the *Dnah5*−/− mouse shows situs inversus, indicating a reversal in the position of internal organs such as the heart, spleen (white arrow), and liver. Each mouse was photographed on 5 mm grid paper. (**E**) Coronal sections stained with HE from wild-type and *Dnah5*−/− mice were compared to assess the size of the lateral ventricle anterior horn with age. The size of the ventricles progressively increased day by day. Scale bars = 500 μm.

**Figure 2 cells-13-01882-f002:**
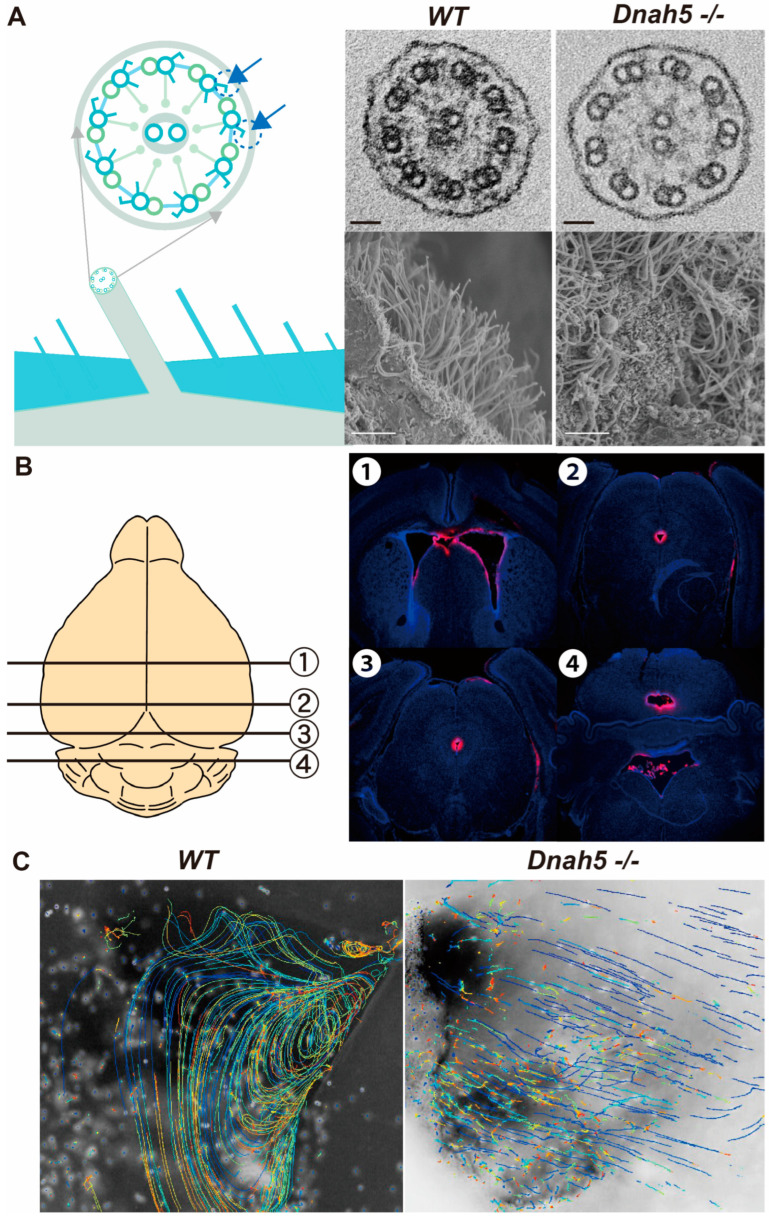
Ciliary structure and CSF flow. (**A**) Observation of cilia under electron microscopy in mice on day three. In the TEM (top, scale bar = 20 nm), a horizontal section of the cilia can be observed. The outer dynein arm structures of the peripheral doublet microtubules, indicated by arrows in the illustration, were absent in *Dnah5*−/− mice, but not in wild-type mice. In the SEM (bottom, scale bar = 5 μm), the ventricular wall observed in *Dnah5*−/− mice exhibited irregular ciliary extension compared to the wild-type mice. (**B**) Confirmation of cerebral aqueductal patency in 4-day-old *Dnah5*−/− mice following injection of the DiI fluorescent dye into the lateral ventricle. The numbered slices 1 to 4 in the illustration represent the locations of coronal brain sections, with slice 1 indicating the anterior horn of the lateral ventricle, slices 2 and 3 depicting the cerebral aqueduct, and slice 4 representing the fourth ventricle. DiI reached the fourth ventricle, indicating patency of the cerebral aqueduct. However, simultaneous observation revealed an already enlarged lateral ventricle. (**C**) Following immediate extraction of brains from 10-day-old wild-type and *Dnah5*−/− mice, coronal sections were created, and microbeads were inserted into the ventricles to track their trajectories. In wild-type mice, microbeads exhibited movement with observed vortex flow inside the ventricle. In contrast, no movement of the microbeads was observed in *Dnah5*−/− mice. The variation in color of the spots in the tracked trajectories represents Z-axis positions, enabling visual evaluation of how the particles move.

**Figure 3 cells-13-01882-f003:**
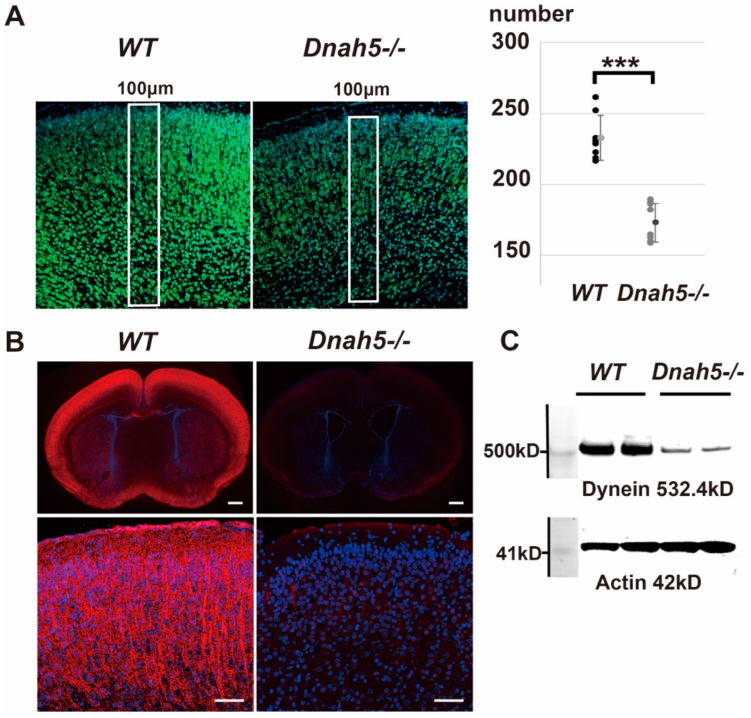
Effects of *Dnah5* gene deficiency on cortex. (**A**) Immunostaining using Neuro Trace (green) and Hoechst 33342 (blue) was performed on the entire six layers of cerebral cortex (100 μm width) within the motor area of the frontal lobe in the brains of 3-day-old wild-type and *Dnah5*−/− mice (left). Cell counts (*n* = 8) revealed a reduction of approximately 25% in the number of neurons in the *Dnah5*−/− mouse cortex compared to the wild-type mouse cortex (right graph, *** *p* < 0.001). (**B**) Staining with anti-dynein antibody (red) and Hoechst 33342 (blue) was performed on wild-type and *Dnah5*−/− mice, capturing images at weak, moderate, and strong magnifications. In wild-type mice, the anti-dynein antibody distinctly stained the cytoplasmic region corresponding to the neuronal axons in the cerebral cortex. Conversely, staining was lower in the *Dnah5*−/− mouse. Top panels, scale bar = 500 μm; bottom panels, scale bar = 50 μm. (**C**) Western blotting confirmed a 56.3% reduction in dynein protein expression in the cerebral cortex of *Dnah5*−/− mice compared to wild-type mice.

**Figure 4 cells-13-01882-f004:**
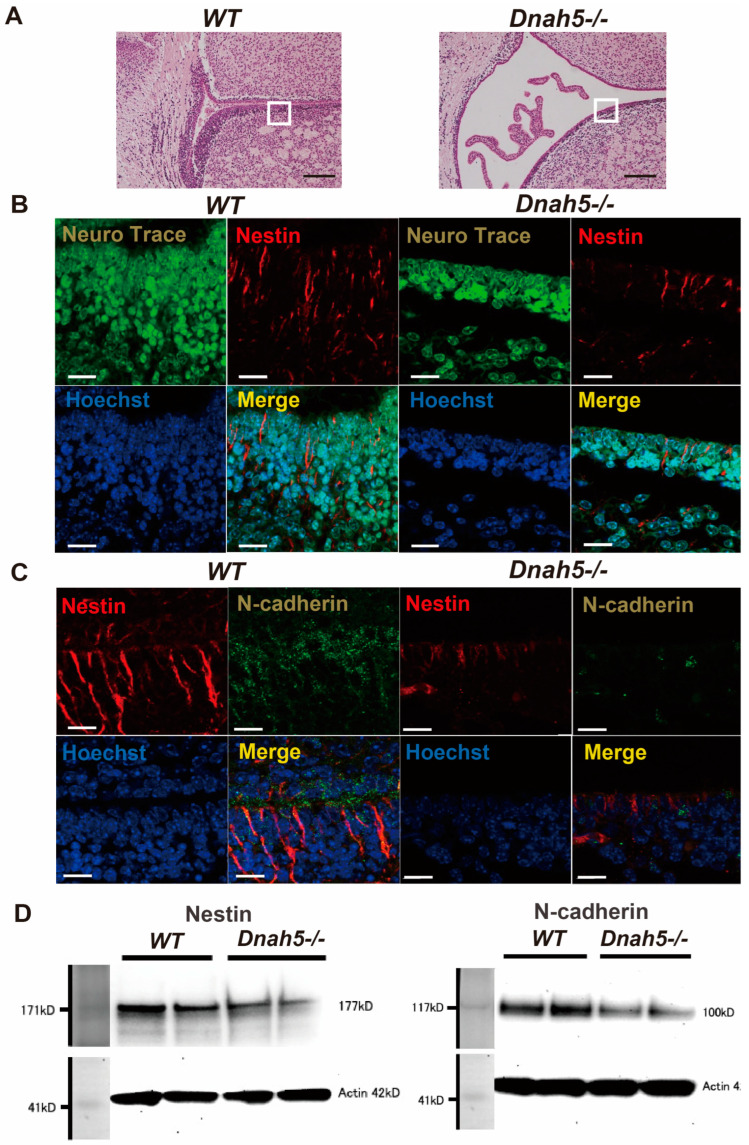
Effects on the ependymal and subependymal zone layers. (**A**) Comparison of the structures of the ventricular wall (ependymal layer and SVZ) between wild-type and *Dnah5*−/− mice in the brain ventricular wall, as indicated by the white boxes in the figure. Scale bar = 200 μm. (**B**) Double immunostaining was performed using Neuro Trace and nestin. Neuro Trace staining revealed a decrease in cell density in the ventricular wall of *Dnah5*−/− mice compared to wild-type mice. Nestin staining also exhibited reduced intensity in *Dnah5*−/− mice. Scale bar = 20 μm. (**C**) Double immunostaining was conducted with N-cadherin and nestin. Similarly to nestin staining, N-cadherin staining showed a reduced intensity in *Dnah5*−/− mice. (Scale bar = 20 μm) (**D**) Western blotting confirmed a 31.7% decrease in nestin protein expression and a 33.3% reduction in N-cadherin protein expression in *Dnah5*−/− mice compared to wild-type mice.

**Table 1 cells-13-01882-t001:** Results of the category analysis using IPA.

	Disease or Function Annotation	*p*-Value	Molecules
1	Development of neural cells	5.82 × 10^−47^	198
2	Development of neurons	4.24 × 10^−45^	189
3	Morphology of nervous system	3.35 × 10^−39^	203
4	Differentiation of neural cells	4.02 × 10^−37^	130
5	Locomotion	2.23 × 10^−36^	105
6	Differentiation of neurons	4.31 × 10^−35^	110
7	Abnormal morphology of nervous system	4.44 × 10^−34^	170
8	Development of head	1.61 × 10^−33^	194
9	Development of body axis	3.48 × 10^−32^	199
10	Morphology of central nervous system	2.05 × 10^−31^	133

**Table 2 cells-13-01882-t002:** Significantly downregulated genes in three-day-old *Dnah5*−/− mice.

Gene Symbols	Description	Fold Change	Chromosome
*NeuroD6*	Neurogenic differentiation 6	−187.65	chr6
*Fezf2*	Fez family zinc finger 2	−62.42	chr14
*Zbtb18*	Zinc finger and BTB domain containing 18	−38.09	chr1
*Mef2c*	Myocyte enhancer factor 2C	−32.05	chr13
*Ptk2b*	PTK2 protein tyrosine kinase 2 beta	−27.68	chr14
*Tbr1*	T-box brain gene 1	−21.66	chr2
*Foxg1*	Forkhead box G1	−20.28	chr12
*Tiam2*	T cell lymphoma invasion and metastasis 2	−18.15	chr17
*Met*	Met proto-oncogene	−17.48	chr6
*NeuroD2*	Neurogenic differentiation 2	−10.44	chr11

**Table 3 cells-13-01882-t003:** Significantly downregulated genes involved in Dynein structure.

Gene	Description	Fold Change
*Dync1h1*	Dynein cytoplasmic 1 heavy chain 1	−2.36
*Dync1i1*	Dynein cytoplasmic 1 intermediate chain 1	−2.58
*Dync1i2*	Dynein cytoplasmic 1 intermediate chain 1	−1.12
*Dync1li1*	Dynein cytoplasmic 1 light intermediate chain 1	1.1
*Dync1li2*	Dynein cytoplasmic 1 light intermediate chain 1	1.19
*Dync2h1*	Dynein cytoplasmic 1 heavy chain 1	−1.26
*Dync2li1*	Dynein cytoplasmic 1 intermediate chain 1	−1.07

## Data Availability

Data are available from the corresponding author upon reasonable request.

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
