# Peer review of "Loss of Dnah5 Downregulates Dync1h1 Expression, Causing Cortical Development Disorders and Congenital Hydrocephalus"

_cells, 2024, doi:10.3390/cells13221882_

Round 1

Reviewer 1 Report

Comments and Suggestions for Authors

The authors study hydrocephalus in mice induced by Dnah5 gene knockout and observe that Dnah5 knock out decrease Dync1h1 expression, which leads to cortical development disorders and congenital hydrocephalus. This is an interesting study, however some concerns raised:

1, the Dnah5 KO down-regulate Dync1h1, which is the key event leading to  cortical development disorders and congenital hydrocephalus. Therefore virus injection into Dnah5 KO mice brain to increase Dync1h1 expression should be able to overcome Dnah5 KO induced pathological changes. The authors are requested to provide these new key evidence.

2, if Dync1h1 is the key molecule accounting for Dnah5 KO induced pathological changes, how Dync1h1 down regulation affect Nestin and N-cadherin expression decreased in Dnah5 KO mice? The authors should answer the question. 

Author Response

Comments and Suggestions for Authors

The authors study hydrocephalus in mice induced by Dnah5 gene knockout and observe that Dnah5 knock out decrease Dync1h1 expression, which leads to cortical development disorders and congenital hydrocephalus. This is an interesting study, however some concerns raised:

1, the Dnah5 KO down-regulate Dync1h1, which is the key event leading to cortical development disorders and congenital hydrocephalus. Therefore virus injection into Dnah5 KO mice brain to increase Dync1h1 expression should be able to overcome Dnah5 KO induced pathological changes. The authors are requested to provide these new key evidence.

Response 1

Thank you very much for your valuable comments.

There has been significant recent interest in developing therapies through transfection in genetically modified mice. However, there are two major practical challenges that would make conducting similar experiments in our study highly difficult.

  • If we were to conduct a transfection experiment, we would need to inject the viral vector into the lateral ventricle within the first three days post-birth to prevent the onset of hydrocephalus. This requires separating neonatal mice from their mother, and upon reintroduction after injection, there is a high probability that the mother would reject the pups, leading to a high risk of experimental failure.
  • Additionally, since Dnah5 KO mice are generated by heterozygous crossbreeding, it is impossible to confirm homozygosity via tail sampling before the injection of the viral vector.

For these reasons, we consider it extremely challenging to conduct transfection experiments on neonatal mice.

Your valuable feedback has prompted me to recognize the importance of further discussing potential applications for future clinical treatments. This insight has made us to broaden our perspective and include new considerations on potential approaches for treating congenital hydrocephalus. I am sincerely grateful for your advice, which has greatly enhanced our study.

2, if Dync1h1 is the key molecule accounting for Dnah5 KO induced pathological changes, how Dync1h1 down regulation affect Nestin and N-cadherin expression decreased in Dnah5 KO mice? The authors should answer the question.

Response 2
The downregulation of Dync1h1 is likely critical in reducing Nestin and N-cadherin expression in Dnah5 KO mice. Dync1h1 encodes a cytoplasmic dynein heavy chain essential for intracellular transport, cellular structure maintenance, and neuron migration. Therefore, a reduction in Dync1h1 expression compromises dynein function, impacting cytoskeletal formation, and ultimately neural progenitor cell development and maintenance.

The following mechanisms are plausible:

  • Maintenance of Neural Progenitor Cells: When Dync1h1 expression is reduced, intracellular transport is impaired, leading to incomplete cytoskeletal construction, which may contribute to decreased Nestin (a neural stem cell marker) expression. Nestin provides structural support necessary for neural progenitor cell maintenance and division, so a reduction in Nestin is directly associated with decreased and delayed development of these cells.
  • Cell Adhesion and Migration: N-cadherin is a protein involved in neuronal migration and adhesion, and Dync1h1 downregulation disrupts N-cadherin localization and expression. This reduction in N-cadherin weakens cell adhesion and inhibits proper migration of neural progenitor cells, potentially disrupting normal cortical layer formation.

Thus, Dync1h1 reduction indirectly contributes to decreased Nestin and N-cadherin expression, correlating with cortical developmental abnormalities and neural progenitor cell loss caused by Dnah5 KO. This mechanism may increase the structural and functional deficits observed in the cortex of Dnah5 KO mice, elevating the risk of congenital hydrocephalus.

Your comments made me realize the need for a more comprehensive discussion on this point. I have expanded our analysis to further elaborate on the relationship between primary cilia, cytoplasmic dynein, and both Nestin and N-cadherin and expect that to improve the manuscript.

Reviewer 2 Report

Comments and Suggestions for Authors

This study provides an intriguing exploration into the role of Dnah5 in the development of hydrocephalus and cortical malformation, specifically through its impact on dynein function and neurogenesis. As a neurosurgeon, I find the use of CRISPR/Cas9 to generate Dnah5 knockout mice particularly valuable, as it allows for a detailed dissection of the genetic and cellular mechanisms involved in congenital hydrocephalus. The reduction in Dync1h1 expression and its correlation with disrupted cortical neuron development presents a compelling case for considering genetic factors in patients with similar cortical abnormalities. However, the paper could benefit from more in-depth discussion on how these findings translate into potential therapeutic strategies for human congenital hydrocephalus, particularly as this condition often presents with complex and multifactorial etiologies.

Additionally, while the study successfully links dynein deficiencies to cortical malformation, the limited focus on primary cilia function raises questions about the broader implications for neuronal migration and CSF dynamics. Addressing these aspects would provide a more comprehensive understanding of the interplay between ciliary dysfunction and cortical development, which is crucial for advancing clinical approaches to hydrocephalus treatment.

Author Response

Reviewer2

Comments and Suggestions for Authors

This study provides an intriguing exploration into the role of Dnah5 in the development of hydrocephalus and cortical malformation, specifically through its impact on dynein function and neurogenesis. As a neurosurgeon, I find the use of CRISPR/Cas9 to generate Dnah5 knockout mice particularly valuable, as it allows for a detailed dissection of the genetic and cellular mechanisms involved in congenital hydrocephalus.

The reduction in Dync1h1 expression and its correlation with disrupted cortical neuron development presents a compelling case for considering genetic factors in patients with similar cortical abnormalities. However, the paper could benefit from more in-depth discussion on how these findings translate into potential therapeutic strategies for human congenital hydrocephalus, particularly as this condition often presents with complex and multifactorial etiologies.

Thank you very much for your valuable feedback.

Response 1
This study offers several important contributions to the therapeutic strategies for congenital hydrocephalus. Specifically, elucidating the mechanism of hydrocephalus development via Dnah5 knockout provides insights that could aid in developing new therapeutic approaches and targets, in addition to conventional treatments. The key contributions are as follows:

  1. Dync1h1 and Dynein Function as Therapeutic Targets
    The study demonstrates that downregulation of Dync1h1 due to Dnah5 deficiency is associated with cortical malformations and hydrocephalus progression. This finding suggests that molecular or gene therapies targeting Dync1h1 or cytoplasmic dynein could be promising approaches. For example, therapies that promote Dync1h1 expression or enhance dynein function may support cortical development and prevent the depletion of neural progenitor cells.
  2. Restoring CSF Flow by Improving Ciliary Motility
    The study shows that impaired ciliary motility induces CSF flow stasis, which exacerbates hydrocephalus. This opens up the possibility of therapeutic strategies aimed at restoring ciliary function, such as pharmacological agents or gene-editing technologies that enhance ciliary motility, which may help mitigate the progression of hydrocephalus.
  3. Early Diagnosis and Prevention of Cortical Development Disorders
    The study suggests that abnormal expression of Dnah5 or Dync1h1 in early fetal stages can lead to cortical development abnormalities, positioning these genes as potential diagnostic markers. Early diagnosis could enable pre- or early postnatal therapeutic intervention, potentially helping prevent hydrocephalus and neurodevelopmental disorders.
  4. Foundation for Personalized Medicine
    This research highlights that congenital hydrocephalus does not rely on a single mechanism but is influenced by multiple intertwined factors. This insight can contribute to the foundation of personalized medicine based on the genetic and symptomatic profiles of individual patients, enabling the design of tailored treatments for specific genetic mutations.

In summary, by clarifying how Dnah5 and Dync1h1 contribute to the etiology and progression of congenital hydrocephalus, this study paves the way for identifying new therapeutic targets, developing early diagnostic technologies, and building a framework for more effective personalized medicine.

Reviewer2

Additionally, while the study successfully links dynein deficiencies to cortical malformation, the limited focus on primary cilia function raises questions about the broader implications for neuronal migration and CSF dynamics. Addressing these aspects would provide a more comprehensive understanding of the interplay between ciliary dysfunction and cortical development, which is crucial for advancing clinical approaches to hydrocephalus treatment.

Response 2

Primary cilia serve as sensory organelles essential for neuronal development and function, exerting broad influences on neuronal migration and cerebrospinal fluid (CSF) dynamics. Below is an overview of the primary roles that primary cilia play in neuronal migration and CSF flow.

  1. Role of Primary Cilia in Neuronal Migration
    Primary cilia are deeply involved in the migration, differentiation, and polarity formation of neurons.
    • Guidance of Neuronal Migration
      Primary cilia detect molecular signals that guide neuron migration, adjusting intracellular signaling pathways accordingly. When primary cilia malfunction, neurons may fail to reach their intended locations, increasing the likelihood of cortical malformations.
    • Signaling Hub
      Primary cilia sense signaling molecules such as Sonic Hedgehog (Shh) and Wnt, facilitating gene expression adjustments within the cell. This regulation affects neuronal proliferation, differentiation, and migration. Malfunction of primary cilia disrupts these critical signaling pathways, potentially resulting in neuronal migration disorders and developmental disabilities.
  2. Impact on CSF Dynamics
    Primary cilia also regulate the flow and dynamics of CSF.
    • Formation of Signal Gradients
      Primary cilia detect growth factors and chemical signals in CSF, guiding the differentiation of neurons and glial cells based on concentration gradients. Normal CSF circulation helps maintain these gradients, and CSF stasis can lead to developmental disorders.
    • Indirect Regulation of CSF Circulation
      While primary cilia do not directly propel CSF, they sense its flow, transmitting information to surrounding cells and modulating the expression of molecules that control CSF dynamics. If primary cilia are deficient, the balance of CSF components is disrupted, adversely affecting neural tissue development and function.
  3. Broad Implications and Links to Congenital Disorders
    Primary cilia dysfunction is associated not only with neural development and CSF circulation but also with congenital disorders.
    • Various Neurodevelopmental Disorders
      Abnormalities in primary cilia-related genes have been linked to brain malformations (e.g., microcephaly, cortical dysplasia) and neurodevelopmental disorders (e.g., autism, epilepsy). Since primary cilia influence both cell migration and CSF flow, their dysfunction can lead to widespread developmental abnormalities.
    • Ciliopathy-Related Disorders (e.g., Joubert Syndrome, Meckel-Gruber Syndrome)
      These disorders stem from primary cilia abnormalities, often presenting with impaired CSF dynamics and cortical development abnormalities. These syndromes also cause issues with brain morphology, laterality, and additional organ abnormalities, further underscoring the importance of primary cilia.

When primary cilia do not function properly, there is a cascade of effects on neuronal migration, CSF flow, and overall neurodevelopment, which may contribute to congenital neurodevelopmental disorders and hydrocephalus.

Your feedback indicated me that my manuscript was indeed lacking certain explanations. I have now added the section regarding potential applications for therapeutic strategies in treating congenital hydrocephalus, as you suggested. Additionally, I have expanded the discussion to include the effects of primary cilia on neuronal migration and CSF dynamics, which were previously insufficiently addressed in the text. Thank you very much for your advice.

Reviewer 3 Report

Comments and Suggestions for Authors

I must say that it aligns very well with the theme we are currently focusing on. The content is relevant and timely, addressing a significant area of research within our field. The presentation of your work is commendable, employing a variety of methods to convey your findings. The inclusion of gene knockout mice, in particular, adds a valuable dimension to your study and enhances the overall impact of your research. Your English writing is averagely clear . It is evident that you have put considerable effort into ensuring that the text is accessible to an international audience.However, there are a few areas where I believe your manuscript could be further strengthened. Specifically, in the sections dealing with immunohistochemistry, Western blot, and immunofluorescence, I would recommend increasing the number of samples and presenting more representative images.  This would not only bolster the statistical significance of your findings but also provide a more comprehensive view of the data. Additionally, I suggest incorporating a simple flowchart outlining the overall thought process of your article. This would help readers to follow the progression of your research more easily. Furthermore, including a schematic diagram illustrating the molecular signaling pathways you have investigated would be beneficial.

Author Response

Reviewer3

Comments and Suggestions for Authors

I must say that it aligns very well with the theme we are currently focusing on. The content is relevant and timely, addressing a significant area of research within our field. The presentation of your work is commendable, employing a variety of methods to convey your findings. The inclusion of gene knockout mice, in particular, adds a valuable dimension to your study and enhances the overall impact of your research. Your English writing is averagely clear. It is evident that you have put considerable effort into ensuring that the text is accessible to an international audience. However, there are a few areas where I believe your manuscript could be further strengthened. â‘ Specifically, in the sections dealing with immunohistochemistry, Western blot, and immunofluorescence, I would recommend increasing the number of samples and presenting more representative images.  This would not only bolster the statistical significance of your findings but also provide a more comprehensive view of the data.

Response1

Thank you for your valuable feedback.

Regarding the sections on immunohistochemistry, Western blot, and immunofluorescence, we have conducted multiple experiments to select the best representative images to present in the manuscript. We have reviewed again all images and found that the present are the best qualitative expression. Regrettably conducting additional experiments may require several months to complete, in fact completing the quite full timeline of the experiment as mentioned below.

If you could provide specific points for improvement, such as particular aspects of the images that require modification, we would be grateful for your guidance.

Thank you once again for your comments.

Reviewer3

â‘¡ Additionally, I suggest incorporating a simple flowchart outlining the overall thought process of your article. This would help readers to follow the progression of your research more easily.  

Response2

Thank you for your valuable feedback.

As you pointed out, the methodology is complex and may be difficult to follow. Therefore, I am in the process of creating a flowchart to clarify the steps involved, as well as a timeline diagram to illustrate the experimental procedures. I will present the data as soon as these visual aids are completed.

Reviewer3

â‘¢ Furthermore, including a schematic diagram illustrating the molecular signaling pathways you have investigated would be beneficial.

Response3

Thank you very much for your valuable feedback.

I would like to further analyze the data obtained from this experiment to illustrate the genetic mutations within the molecular signaling pathways. However, as your suggestion was made relatively recently, it may be challenging to meet the November 4th deadline. I kindly request a little more time to complete this work. Thank you for your understanding.

Round 2

Reviewer 1 Report

Comments and Suggestions for Authors

The authors have answered all my concerns properly and significantly improve their works with more discussions. I suggest accepting the manuscript in the current form.